# Characterization of NDM-5-Producing *Escherichia coli* Strains Isolated from Pediatric Patients with Bloodstream Infections in a Chinese Hospital

**DOI:** 10.3390/genes14020520

**Published:** 2023-02-18

**Authors:** Lili Huang, Hongye Hu, Chen Xu, Mi Zhou, Yuanyuan Li, Yunbing Li, Shuyan Wu, Ning Dong

**Affiliations:** 1Laboratory Department, Children’s Hospital of Soochow University, Suzhou 215025, China; 2Department of Medical Microbiology, School of Biology and Basic Medical Sciences, Medical College, Soochow University, Suzhou 215127, China; 3Department of Laboratory Medicine, School of Medicine, Jiangsu University, Zhenjiang 212013, China; 4Department of Pharmacy, Children’s Hospital of Soochow University, Suzhou 215025, China; 5Department of Medical Microbiology, Experimental Center, Medical College, Soochow University, Suzhou 215127, China; 6Suzhou Key Laboratory of Pathogen Bioscience and Anti-infective Medicine, School of Biology and Basic Medical Sciences, Soochow University, Suzhou 215127, China

**Keywords:** NDM-5, carbapenemase, *Escherichia coli*, multidrug resistance, bloodstream infections, whole-genome sequencing

## Abstract

*Escherichia coli* (*E. coli*) bloodstream infections (BSIs) are among the most predominant causes of death in infants and children worldwide. NDM-5 (New Delhi Metallo-lactamase-5) is responsible for one of the main mechanisms of carbapenem resistance in *E. coli*. To analyze the phenotypic and genomic characteristics of NDM-5-producing *E. coli* from bloodstream infections (BSIs), a total of 114 *E. coli* strains was collected from a children’s hospital in Jiangsu province, China. Eight *bla*_NDM-5_-carrying *E. coli* strains were identified which were all carbapenem-resistant and carried diverse antimicrobial resistance genes apart from *bla*_NDM-5_. They belonged to six distinct sequence types (STs) and serotypes including one each for ST38/O7:H8, ST58/O?:H37, ST131/O25:H4, ST156/O11:H25 and ST361/O9:H30 and three strains are originating from a single clone belonging to ST410/O?:H9. Apart from *bla*_NDM-5_, the *E. coli* strains isolated from BSIs also carried other β-lactamase genes, including *bla*_CMY-2_ (*n* = 4), *bla*_CTX-M-14_ (*n* = 2), *bla*_CTX-M-15_ (*n* = 3), *bla*_CTX-M-65_ (*n* = 1), *bla*_OXA-1_ (*n* = 4) and *bla*_TEM-1B_ (*n* = 5). The *bla*_NDM-5_ genes were located on three different types of plasmids, which were IncFII/I1 (*n* = 1), IncX3 (*n* = 4) and IncFIA/FIB/FII/Q1 (*n* = 3). The former two types were conjugatively transferable at frequencies of 10^−3^ and 10^−6^, respectively. The dissemination of NDM-producing strains, which exhibit resistance to the last-line antibiotics, carbapenems, may increase the muti-antimicrobial resistance burden among *E. coli* BSIs and further threaten public health.

## 1. Introduction

Bloodstream infections (BSIs) are considered among the most serious nosocomial pathologies with limited choices of effective therapeutic options. BSIs are frequently associated with high morbidity and mortality in hospitalized patients, which increases treatment costs, patients’ burden and diagnostic uncertainties [1]. According to the published data, 48.9 million incident cases have taken place worldwide, and 11.0 million deaths were associated with BSIs, which represented 19.7% of all global deaths [2]. Of note, BSIs remain a major problem for neonates and young children [3,4], because the diagnosis of BSIs in children is relatively difficult as they could manifest as serious infections in other sites of the body. Neonatal BSIs are still among the major reasons of morbidity and mortality in spite of the remarkable progress in neonatal medicine [5]. It was estimated that 3 million neonates were affected by BSIs, with mortality between 11% and 19%, worldwide [6].

BSIs are increasing globally, and in addition, the causative organisms are becoming resistant to clinical front-line antibiotics, which increases in-hospital mortality and length of stay [7]. BSIs are identified by positive blood cultures in patients with systemic signs of infection such as acute organ dysfunction, respiratory failure and even septic shock [8]. It was reported that sepsis was responsible for 60–80% of lost lives per year in children [9]. With rapid evolution and the acquisition of antibiotic resistance, Gram-negative strains are greatly responsible for BSIs [10]. According to previous studies, BSIs caused by MDR Enterobacterales increased significantly from 6.2% to 15.8% from 1997 to 2016 [11]. *Escherichia coli* was among the most common pathogens responsible for nosocomial BSIs [12,13,14]. In addition, *E. coli* BSIs are among the most predominant cause of death in infants and children worldwide [15]. The incidence of *E. coli* BSIs has significantly increased in England [16] and Australia [17] in recent years. A systematic review on sub-Saharan Africa found that *E. coli* accounted for 10% of blood culture positive for bacteriaemia or sepsis [18]. Meanwhile, BSIs are common among children cancer patients under five years old, and the most identified microorganisms in Egypt were Gram-negative bacteria, with *E. coli* being the dominant species (27.8%) [19]. Early-onset BSIs caused by *E. coli* are potentially fatal diseases (36.6%) among preterm infants [20].

As a member of the ESKAPE group of pathogens, *E. coli* strains that contain extended-spectrum β-lactamases (ESBLs) and carbapenemases could usually develop resistance to a variety of antimicrobial agents [21,22,23]. Resistance to third-generation cephalosporin and carbapenems in *E. coli* bloodstream isolates has rapidly increased [24,25]. ESBLs-producing Enterobacteriaceae (especially, *E. coli*) are important human pathogens and a cause of bloodstream infections that appeared multidrug resistant in developed and developing countries [26]. *E. coli* strains encoding CTX-M-β-lactamases are considered the widely distributed species relevant to globally disseminated ESBLs [27]. To date, 220 different variants of CTX-M-β-lactamases have been identified and clustered into five subfamilies based on their amino acid residues [28]. Recent molecular epidemiology studies in China have showed that CTX-M-14, CTX-M-15 and CTX-M-55 were the most common ESBLs; according to the Multi-Locus Sequence Typing (MLST) analysis, these ESBLs were often found in the sequence types ST131, ST405 and ST69 [29]. The first Indian report on a genome-wide comparison displayed the distribution of *bla*_NDM-5_ among ST131, ST405 and ST410, whereas *bla*_CTX-M-15_ was associated with cephalosporin resistance in ST131 and ST405 [30]. Whole-genome analysis in Australia, New Zealand and Singapore showed that Clade C1/C2 (30.2% and 53.5% of ST131, respectively) carrying CTX-M-type (CTX-M-14, CTX-M-15 (mainly) and CTX-M-27) ESBLs was clearly predominant [31]. The ST131 and ST410 clones are important human *E. coli* lineage among extraintestinal pathogenic *E. coli* (ExPEC) isolates worldwide [32,33]. The majority of ST131 isolates belong to the O25b:H4 serotype; other isolates in some clinical collections appeared to belong to the serotype O16:H5 [34]. Unlike ST131, ST410 has a wide distribution among different populations (humans, animals and livestock) [35]. The ST131 and ST410 clones are frequently resistant to most β-lactams, aminoglycosides and fluoroquinolones and could cause a great variety of infections, such as bloodstream infections, intra-abdominal infections and septic shock.

Carbapenems are the ‘last-resort’ antibiotics for treating severe infections caused by multidrug-resistant bacteria, yet the acquisition of carbapenemase genes by Gram-negative strains, particularly by Enterobacteriaceae, significantly compromises their efficacy [36]. Accordingly, about 32% of patients died within 14 days from bloodstream infections caused by carbapenem-resistant Enterobacteriaceae (CRE) [25]. The carbapenemase genes are often located on mobile genetic elements such as plasmids, and horizontal gene transfer can accelerate the spread of CRE between species worldwide [37]. Since research in 2008 showed that a novel carbapenem resistance gene, NDM , a carbapenem-hydrolyzing enzyme, was found in *Klebsiella pneumoniae* isolated from a patient in India, this gene has disseminated over many countries [38]. NDM-type carbapenemases mediate resistance to most β-lactams antibiotic except aztreonam [39]. NDM variants are increasingly becoming one of the main effectors of carbapenem resistance among *E. coli* isolates in clinical settings in China [40]. NDM-positive *E. coli* isolated from BSIs were reported from different countries across the world, including India [41], China [42], Sweden [43], Cuba [44] and several Latin American countries [45]. Despite their prevalence being relatively low, such strains could cause considerable mortality and pose great challenges to the treatment of bloodstream infections [46].

At the time of writing, a total of 44 NDM variants, namely, NDM-1 to NDM-44, were deposited in the database (https://www.ncbi.nlm.nih.gov/pathogens/refgene/#gene_family: (*bla*_NDM_) (accessed on 1 November 2022)). Among these, NDM-5 was firstly detected in *E. coli* isolated from the throat of a patient with a history of travel to India in 2011 [47] and subsequently reported in different countries including India [48], South Korea [49], Australia [50], America [51] and Egypt [52]. NDM-5 was also reported to be the dominant NDM variant produced by *E. coli* strains from different hospitals in Jiangsu province, China [53,54].

Yet, the phenotypic and genetic characteristics of the clinical NDM-5-producing *E. coli* strains remain largely unknown. To fill this gap, we collected a total of eight NDM-5-producing *E. coli* strains isolated from between years 2016 and 2020 from BSIs in a children’s hospital in Jiangsu, China. The clinical information of the studied strains was also collected. The antimicrobial resistance phenotypes and the genetic characteristics of all strains were comprehensively studied. In this study, we conducted a bioinformatics analysis to determine the molecular characteristics of NDM-5-positive *E. coli* carried by pediatric patients and assessed the transmission ability of the plasmid harboring *bla*_NDM-5_. Our findings shall increase the understanding of how to control of untreatable BSIs caused by carbapenem-resistant *E. coli*.

## 2. Materials and Methods

### 2.1. Bacteria Isolates and Clinical Data Collection

A total of 114 non-duplicated *E. coli* strains isolated from blood specimens from 114 patients from the Children’s Hospital of Soochow University (Suzhou, China) were collected during the period from September 2016 to October 2020. The total numbers of *E. coli* isolates during these periods were 1 in 2016 and 2019, respectively, 3 in 2017 and 3 in 2020. The strains were isolated in accordance with standard procedures. Briefly, the blood samples were first cultured with the BD BACTEC^TM^ FX Blood Culture System (BD Diagnostics, Sparks, MD, USA). The positive cultures were sub-cultured on blood agar and in Mueller–Hinton broth at 37 °C for 16–24 h to obtain the *E. coli* isolates. All *E. coli* isolates from patients meeting the inclusion criteria of the study were stored in the laboratory at –80 °C in cryovials containing 20% glycerol and the nutrient broth for further analysis. The clinical information of the patients from whom the *bla*_NDM-5_-positive *E. coli* isolates were isolated was obtained from the clinicians. The relevant clinical information and the origin of the strains are presented in Table 1.

### 2.2. Strain Identification and β-Lactamase Genes Confirmation

The species of all strains were identified using MALDI-TOF MS (Bruker Daltonik GmbH, Bremen, Germany) and 16S rRNA gene sequencing. The presence of the β-lactamase genes, including carbapenemase (*bla*_NDM-5_) and ESBLs genes (*bla*_CMY-2_, *bla*_CTX-M-15_, *bla*_OXA-1,_ and *bla*_TEM-1B_) was detected by PCR targeting the entire *bla*_NDM_ gene and other β-lactamase genes, which were validated with Sanger sequencing [55].

### 2.3. Antimicrobial Susceptibility Testing

Antimicrobial susceptibility testing was conducted for all isolates against 14 commonly used antibiotics including extended-spectrum β-lactam (ampicillin), aminoglycoside (gentamicin, tobramycin), monobactam (aztreonam), β-lactam/β-lactamase inhibitor combinations (ampicillin–sulbactam, piperacillin–tazobactam), carbapenem (imipenem), cephalosporins (cefepime, ceftriaxone, ceftazidime, cefotetan), fluoroquinolones (ciprofloxacin, levofloxacin) and trimethoprim–sulfamethoxazole. Antimicrobial susceptibility testing was conducted using the micro-broth dilution method. The minimum inhibitory concentrations (MICs) were interpreted according to the Clinical and Laboratory Standards Institute (CLSI) guidelines [56]. *E. coli* ATCC 25922 was used as a quality control strain.

### 2.4. Conjugation Assay

A conjugation assay was performed using the filter-mating method [57]. Briefly, each *bla*_NDM-5_-carrying *E. coli* strain was used as the donor, and rifampicin-resistant *E. coli* EC600 was used as the recipient to test the transferability of the carbapenem resistance gene, *bla*_NDM-5_. The donor and the recipient were mixed in the ratio of 1:1 and cultured overnight on a nitrocellulose filter (HA type; pore size, 0.22 µm; Millipore Corp, Billerica, MA, USA) placed on LB agar. The filters were removed from the plates and placed in 1.0 mL of phosphate buffer. The cells were removed from the filter with a Vortex mixer. Dilutions were made, and all transconjugants were selected on LB agar plates supplemented with 1 µg/mL of meropenem and 600 µg/mL of rifampicin. The transconjugants were confirmed by PCR targeting the *bla*_NDM-5_ gene and by antimicrobial susceptibility testing. The conjugation frequencies were estimated by dividing the number of colony-forming units (CFUs) of the transconjugants by the number of CFUs of the recipient strains *E. coli* EC600.

### 2.5. DNA Extraction

The genomic DNA of each *bla*_NDM-5_-carrying *E. coli* was extracted from overnight cultures on BHI using a Genomic DNA Extraction Kit (QIAGEN, Valencia, CA, USA) according to the manufacturer’s instructions. The purity and concentrations of DNA were measured by a NanoDrop spectrophotometer (Thermo Scientific, Waltham, MA, USA).

### 2.6. Whole-Genome Sequencing and Bioinformatics Analysis

The extracted genomic DNA was subjected to whole-genome sequencing using the HiSeq platform (Illumina, San Diego, CA, USA). Trimmomatic was used to trim the raw reads in order to remove the adaptors and low-quality sequences [58]. De novo genome assembly was conducted with SPAdes v3.15.5 [59]. The draft genome sequences were annotated using RAST [60]. Acquired antimicrobial resistance genes (ARGs) were identified with ResFinder 4.1 [61]. Insertion sequences (ISs) were identified with the online software ISfinder [62]. Plasmid replicon types were identified with PlasmidFinder [63]. The comparison of the plasmids was performed using BLASTn, and the results were analyzed with the R tool [64]. The O:H serotype was identified with ECTyper v0.8.1 [65]. MLST v2.11 was used to determine the variety of STs [66]. Variant calling and core genome alignment of the strains E3, E4 and E5 was conducted using Snippy 3.1 [67]. Pairwise SNP was calculated using snp-dists 0.8.2 [68]. The harvest suite v.1.2 was used to remove recombination sequences and conduct the phylogenetic analysis using the assembled genome sequences as the input [69]. The generated phylogenetic tree was visualized and modified with iTOL v6 [70]. A heatmap of antimicrobial resistance genes was obtained using TBtools [71]. Plasmid alignment was visualized with BRIG [72].

### 2.7. Ethical Statements

For the study was obtained through the Children’s Hospital of Soochow University (Suzhou, China).

## 3. Results

### 3.1. Clinical Information and Overview of the NDM-5-Producing E. coli Strains

Among the 114 *E. coli* strains from BSIs collected from 2016 to 2020 in a children’s hospital in China, 8 (7.02%) strains designated E1–E8 carried *bla*_NDM-5_. The age of the eight patients ranged from 9 days to 13 years. Three (37.5%) were male patients, and five (62.5%) were female patients. The patients suffered from different diseases including acute myeloid leukemia, acute lymphoblastic leukemia, neonatal sepsis, congenital atresia of the biliary tract, etc., and were admitted to different departments in the hospital including hematology (*n* = 4), neonatology (*n* = 3) and gastroenterology (*n* = 1). The patients were treated with different antibiotics such as meropenem, tigecycline and amikacin. Seven patients recovered, and one patient was discharged automatically (Table 1).

### 3.2. Antimicrobial Resistance Profiles of the NDM-5-Producing E. coli Strains

The minimum inhibitory concentrations of commonly used antibiotics for the NDM-5-producing *E. coli* strains are listed in Table 2. All these strains exhibited multidrug resistance phenotypes. All eight strains were resistant to ampicillin, imipenem, cefepime, ceftriaxone, ceftazidime, cefotetan, ampicillin–sulbactam, piperacillin–tazobactam and trimethoprim–sulfamethoxazole. The proportion of *E. coli* strains that were non-susceptible to aztreonam, gentamycin and tobramycin were 62.5%, 50% and 75%, respectively. In addition, 87.5% of the eight strains were resistant to ciprofloxacin, and the remaining 12.5% strains exhibited intermediate resistance to ciprofloxacin. We found that 62.5% and 37.5% of the strains were resistant and intermediately resistant to levofloxacin, respectively. 

### 3.3. Genomic Characteristics of the NDM-5-Producing E. coli Strains

Draft genome sequences of the eight NDM-5-producing *E. coli* strains were obtained. These strains belonged to six different sequence types including ST410 (*n* = 3, strains E3, E4 and E5), ST38 (*n* = 1, strain E1), ST58 (*n* = 1, strain E2), ST131 (*n* = 1, strain E6), ST361 (*n* = 1, strain E7) and ST156 (*n* = 1, strain E8). The strains E3, E4 and E5 belonged to an unknown O serotype (O?) and to serotype H9. The strains E1, E2, E6, E7 and E8 belonged to the serotypes O7:H18, O?:H37, O25:H4, O9:H30 and O11:H25, respectively. The strains E3, E4 and E5 shared the same sequence type and serotype, and the number of pairwise SNPs among the three strains was <20, suggesting they belonged to a single clone (Figure 1).

The *E. coli* strains carried different antimicrobial resistance genes conferring resistance to different classes of antibiotics including aminoglycoside, fosfomycin, fluoroquinolone, macrolides, chloramphenicol, rifampicin, sulphonamide, tetracycline, trimethoprim, extended-spectrum β-lactam and carbapenem. The number of acquired antimicrobial resistance genes carried by each *E. coli* strain ranged from 8 to 17. Two strains (E1 and E2) carried the tetracycline resistance gene, *tet*(A), while four strains (E3, E4, E5 and E8) harbored the tetracycline resistance gene *tet*(B). Apart from *bla*_NDM-5_, the *E. coli* strains isolated from BSIs also carried other β-lactamase genes, including *bla*_CMY-2_ (*n* = 4), *bla*_CTX-M-14_ (*n* = 2), *bla*_CTX-M-15_ (*n* = 3), *bla*_CTX-M-65_ (*n* = 1), *bla*_OXA-1_ (*n* = 4) and *bla*_TEM-1B_ (*n* = 5). Particularly, the strains E3, E4 and E5 carried five β-lactamase genes, including *bla*_NDM-5_, *bla*_CMY-2_, *bla*_CTX-M-15_, *bla*_OXA-1_ and *bla*_TEM-1B_, which could pose a significant threat to public health. Despite belonging to a single clone, the resistance genes carried by the three strains were not identical, suggesting active genetic recombination could have occurred in this high-risk clone. The strain E1 carried *bla*_CMY-2_ and *bla*_TEM-1B_, the strain E2 carried *bla*_OXA-1_, the strain E6 carried *bla*_CTX-M-14_ and *bla*_TEM-1B_, _the_ strain E7 carried *bla*_CTX-M-14_ and the strain E8 carried *bla*_CTX-M-65_ (Figure 2).

### 3.4. Transferability and Genetic Analysis of bla_NDM-5_-Carrying Plasmids

To further investigate the role of NDM-5-producing *E. coli* strains in bacterial dissemination, we analyzed the drafts of the whole-genome sequences of all strains and found that the *bla*_NDM-5_ genes were located in contigs ranging from 3 Kb to 43 Kb. The *bla*_NDM-5_ gene in strain E2 was located on a 43,563 bp contig, and the *bla*_NDM-5_ genes in other strains were all located on contigs < 9 Kb. The *bla*_NDM-5_-carrying contigs were generally too short to accurately resolve the genetic context of *bla*_NDM-5_. The *bla*_NDM-5_-bearing contigs in these strains aligned well to different plasmids, including the 107kb IncFII/I1 plasmid pE−T654−NDM-5 (accession no.: CP090291, aligned strain: E1; Figure 3), the 46 kb IncX3 plasmid pL65-9 (accession no.: CP034744; aligned strains: E2, E6, E7 and E8; Figure 4) and the 108 kb IncFIA/FIB/FII/Q1 plasmid pC405−NDM5 (accession no.: LC521844; aligned strains: E3, E4 and E5; Figure 5). The *bla*_NDM-5_ genes in the strains E1, E6, E7 and E8 were conjugatively transferable to *E. coli* EC600 with conjugation efficiencies of 2.5 × 10^−3^, 2.9 × 10^−6^, 3.5 × 10^−6^ and 1.6 × 10^−6^, respectively. The transconjugants were all resistant to carbapenems and cephalosporins (Table 2). *bla*_NDM-5_ in the strains E2, E3, E4 and E5 were non-conjugatively transferable under the experimental conditions of this study.

## 4. Discussion

Carbapenems are used as the last-resort antibiotics to treat Gram-negative bacterial infections. The emergence of carbapenem-resistant organisms is considered one of the most urgent public health concerns [73]. The acquisition of carbapenemase genes by Gram-negative strains is one of the major mechanisms of carbapenem resistance, and the horizontal gene transfer of carbapenemase genes located on mobile genetic elements, including transposons and plasmids, is causing a widespread dissemination of CRE [37]. *E. coli* is among the most frequently isolated CRE worldwide which could colonize both elderly patients (aged >60 years) and the pediatric population [74,75]. Carbapenem-resistant *E. coli* isolated from blood samples has been reported across the world [76]. Carbapenem-resistant *E. coli* strains carrying different *bla*_NDM_ gene variants were found in North America, Africa, Asia and Europe [77]. A study from hospitals in Chongqing, China, on strains from BSIs suggested that the infection ratio of *E. coli* in children was significantly higher than in adults, and most of the *E. coli* strains from BSIs in children expressed *bla*_NDM_, particularly *bla*_NDM-5_ [78]. Furthermore, the spread of ESBL-producing *E. coli* is one of the most important driving factors of the abuse of carbapenems, which indirectly exacerbates the selective pressure of carbapenemase producers [79]. It is well known that CTX-M-15 and CTX-M-14 are the most common CTX-Ms worldwide, often reported in South-East Asia [27]. Our study also demonstrated that carbapenemase-producing *E. coli* are often multidrug resistant and usually express other antibiotic resistance genes.

The genomic and phenotypic characteristics of *bla*_NDM-5_/*bla*_CTX-M_-positive *E. coli* from BSIs still remain poorly understood. In this study, a total of eight NDM-5-producing strains were obtained from 114 *E. coli* strains from BSIs in pediatric patients. The number of NDM-5-producing *E. coli* isolates was higher in 2017 (*n* = 3) and 2020 (*n* = 3) than in 2016 (*n* = 1) and 2019 (*n* = 1). No NDM-5-producing *E. coli* was isolated from blood samples in 2018. This could be associated with the different number of samples collected each year. After treatment with different antibiotics such as meropenem, tigecycline and amikacin, seven patients recovered, and one patient was discharged automatically. The *E. coli* strains causing BSIs carry genes responsible for resistance to aminoglycosides [*strA*, *strB*, *aac(3)-IId*, *aac(6)-Ib-cr*, *aac(3)-IV*, *aadA2*, *aadA5*, *aph(3)-la*, *aph(4)-la*], extended-spectrum β-lactams (*bla*_CMY-2_, *bla*_CTX-M-15_, *bla*_OXA-1_, *bla*_TEM-1B_ and *bla*_NDM-5_) and fluoroquinolones (*qnrS1*, *qnrS2*). These findings are well in accordance with the multidrug resistance to aminoglycosides, third-generation cephalosporins, carbapenems and quinolones determined in this study. A previous study reported the first genome sequence of an MDR *E. coli* carrying *bla*_NDM-5_ and two copies of *bla*_CTX-M-14_ from a BSI in China [80]. Similarly, our research showed that both an ESBL gene and a carbapenemase gene were detected in six isolates, including *bla*_CTX-M-15_/*bla*_NDM-5_ in strains E3, E4 and E5, *bla*_CTX-M-14_/*bla*_NDM-5_ in strains E6 and E7 and *bla*_CTX-M-65_/*bla*_NDM-5_ in strain E8. Additionally, a previous study firstly reported an *E. coli* ST410 strain coharboring the *bla*_NDM-5_, *bla*_OXA-1_, *bla*_CTX-M-15_, *bla*_CMY-2_, *aac(3)-IIa* and *aac(6)-Ib-cr* genes that was collected from a bloodstream infection in China [81]. We also found that an *E. coli* ST410 strain coharbored the *bla*_NDM-5_, *bla*_OXA-1_, *bla*_CTX-M-15_, *bla*_CMY-2_, *bla*_TEM-1D_, *bla*_AmpC1_, *aac(3)-IId* and *aac(6)-Ib-cr* genes; it was collected from a bloodstream infection in Suzhou, China.

The eight NDM-5-producing strains belonged to six different sequence types including ST410, ST38, ST58, ST131, ST361 and ST156, all of which have been reported to be associated with multidrug resistance phenotypes. Among the sequence types in this study, ST410, ST38, ST361 and ST156 *E. coli* were identified as NDM-5-producers in previous studies [53,82,83,84]. ST131 and ST58 *E. coli* have been reported to be NDM-1 producers [85,86]. Previous epidemiological studies demonstrated that the dominant *E. coli* sequence type associated with BSIs in China was ST131, followed by ST69 and ST38 [87,88]. ST131 is the predominant *E. coli* lineage among extraintestinal pathogenic *E. coli* (ExPEC) isolates worldwide and is commonly reported to produce extended-spectrum β-lactamases [32]. Previous studies reported that CTX-M-14 and CTX-M-15 were the most common ESBLs in *E. coli* ST131 isolates [32], which is in line with the results of our study. Two serotypes of ST131, O16:H5 and O25b:H4 have been identified [34]. In 2008, the serogroup O25b and the sequence type ST131were detected in many countries from analyses of the population biology of ESBL-producing *E. coli* [89]. The strain E6 in this study belongs to O25b:H4, ST131. Due to the limited number of strains included in this study, we were unable to compare the prevalence of different sequence types that we calculated with that reported in the literature.

Three strains, belonging to a single clone with the sequence type ST410 and the serotype O?:H9, were collected in this study. ST410 has been reported worldwide as an extraintestinal pathogen associated with resistance to different classes of antibiotics such as fluoroquinolones, polymyxins, third-generation cephalosporins and carbapenems [33,90]. ST410 was further classified into several clades, each of which is associated with a different serotype, including clades A and B (serotype O8:H21), clade C (O8:H9), and clade D and E (O?:H9) [35]. We found that the strains E3, E4 and E5 recovered from bloodstream infection belonged to the serotype clades D and E, which has not been published before. Strains belonging to the clades D and E could be recovered from human and animal samples, posing threat to both populations [35]. The clone ST410 *E. coli* undergoes constant evolution, with its resistome changing with time. Around 2014, a clade of ST410 acquired the carbapenemase gene, *bla*_NDM-5_, on a conserved IncFII plasmid; this could be the ancestral source of the *bla*_NDM-5_-carrying plasmids in the ST410 strains E3, E4 and E5 analyzed in this study [33]. A previous study reported the isolation of an ST410 *E. coli* strain a carrying *bla*_NDM-5_-encoding IncX3 plasmid from a 59-year-old patient with BSI in China, yet the ST410 *E. coli* carrying a *bla*_NDM-5_-encoding IncFII plasmid isolated from blood samples of patients with BSIs was not reported before this study [91]. In addition, we reported the first ST410 clone which co-harbored five β-lactamase genes, including *bla*_NDM-5_, *bla*_CMY-2_, *bla*_CTX-M-15_, *bla*_OXA-1_ and *bla*_TEM-1B_. Considering that the ST410 clone isolated in this study carries a cassette of important resistance genes, has the potential of transmission between patients and could cause BSIs, it should be monitored closely in the future.

Plasmids play a pivotal role in the dissemination of antimicrobial resistance genes. *bla*_NDM-5_ has been reported to be closely associated with different types of plasmids such as IncX3, IncFII and IncHI2 [91,92,93]. Among these, the IncF plasmid, with a low copy number, has a narrow host range (Enterobacteriaceae) and has greatly improved the fitness cost through its antimicrobial resistance determinants by horizontal gene transfer [94]. This is the most common plasmid type in the ST131 clone and is widely distributed in clinically Enterobacteriaceae. The IncX3 plasmid containing *bla*_NDM-5_ has showed a varied geographical distribution in China, as it has been isolated in Jiangsu province [95], Shanghai [96] and Zhejiang province [97]. The *bla*_NDM-5_ genes in the strains E1−E8 were all plasmid-borne. Three different types of *bla*_NDM-5_-carrying plasmids were identified among the strains, including the IncFII/I1 pE−T654−NDM-5−like plasmids, the IncX3 pL65−9−like plasmids and the IncFIA/FIB/FII/Q1 pC405−NDM5−like plasmids. The former two types carry conjugation-associated genes and are conjugatively transferable at different frequencies. The pC405−NDM5−like plasmids are non-self-transmissible. The reason that *bla*_NDM-5_ in the strains E3, E4 and E5 are non-conjugative transferable could be explained by the fact that the IncFIA/FIB/FII/Q1 pC405−NDM5−like plasmid does not carry conjugation-associated genes. The fact that no transconjugants were detected for strain E2 could be explained by a low conjugation frequency (~10^−6^) of the IncX3 plasmids. The transferability of *bla*_NDM-5_-carrying plasmids may promote the rapid dissemination of resistance-encoding elements among Gram-negative bacterial pathogens. Controlled measures such as optimizing antibiotic usage and the development of new antibiotics or antibiotic adjuvants are urgently needed to prevent the emergence and transmission of resistant bacteria such as carbapenem-resistant *E. coli*.

## 5. Conclusions

In summary, this study collected 8 *bla*_NDM-5_-carrying *E. coli* strains from a total of 114 *E. coli* strains from bloodstream infections in a children’s hospital in Jiangsu province, China. These strains were carbapenem-resistant and carried diverse antimicrobial resistance genes apart from *bla*_NDM-5_. The eight strains belonged to six distinct STs and serotypes including one each for ST38/O7:H8, ST58/O?:H37, ST131/O25:H4, ST156/O11:H25 and ST361/O9:H30, and three strains are originating from a single clone belonging to ST410/O?:H9. Five β-lactamase genes were identified in the ST410 *E. coli* strains, i.e., *bla*_NDM-5_, *bla*_CMY-2_, *bla*_CTX-M-15_, *bla*_OXA-1_ and *bla*_TEM-1B_. The *bla*_NDM-5_ gene was located on three different types of plasmids, which were IncFII/I1 (*n* = 1), IncX3 (*n* = 4) and IncFIA/FIB/FII/Q1 (*n* = 3). The former two types were conjugatively transferable at frequencies of 10^−3^ and 10^−6^, respectively. The results of this study could increase our understanding of carbapenem-resistant *E. coli* from BSIs. Effective strategies such as the development of new antibiotics or antibiotic adjuvants and the optimization of antibiotic usage are needed for the control of untreatable BSIs caused by carbapenem-resistant *E. coli*.

## Figures and Tables

**Figure 1 genes-14-00520-f001:**
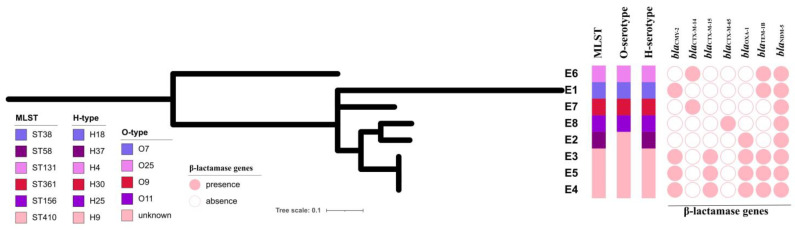
Phylogeny of *bla*_NDM-5_-positive *E. coli* isolates from bloodstream infections. MLST, O and H serotypes and the presence of β-lactamase genes are plotted.

**Figure 2 genes-14-00520-f002:**
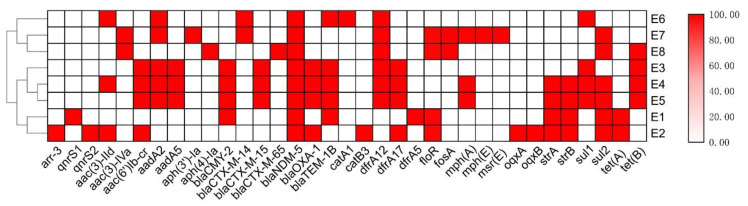
Heatmap of antimicrobial resistance genes carried by *E. coli* isolates in this study. The horizontal axis represents the antimicrobial resistance genes, and the vertical axis represents the strain IDs. The red boxes represent the presence of the corresponding items among the sequenced isolates, and the white boxes represent their absence. The gradient identity bar indicates the percentage similarity of related genes. The similarity tree was constructed using agglomerative hierarchical clustering, with the degree of similarity between different clusters calculated by the average linkage method and the degree of similarity of different isolates (calculated using the Spearman’s rank correlation coefficient).The classes of antibiotics to which the genes confer resistance included aminoglycosides: *strA*, *strB*, *aac(3)-IId*, *aac(6)-Ib-cr*, *aadA2*, *aadA5*, *aph3-Ia*, *aph(4)-Ia* and *aac(3)-IV*; fosfomycin: *fosA*; fluoroquinolones: *qnrS1* and *qnrS2*; macrolides: *mph*(A), *mph*(B), *mph*(E) and *msr*(E); chloramphenicol: *catA1*, *catB3*, and *floR*; rifampicin: *arr-3*; sulphonamide: *sul1* and *sul2*; tetracycline: *tet*(A) and *tet*(B); trimethoprim: *dfrA5*, *dfrA12* and *dfrA17*; extended-spectrum β-lactams: *bla*_CMY-2_, *bla*_TEM-1D_, *bla*_OXA-1_, *bla*_CTX-M-14_, *bla*_CTX-M-15_, and *bla*_CTX-M-65_; carbapenems: *bla*_NDM-5_.

**Figure 3 genes-14-00520-f003:**
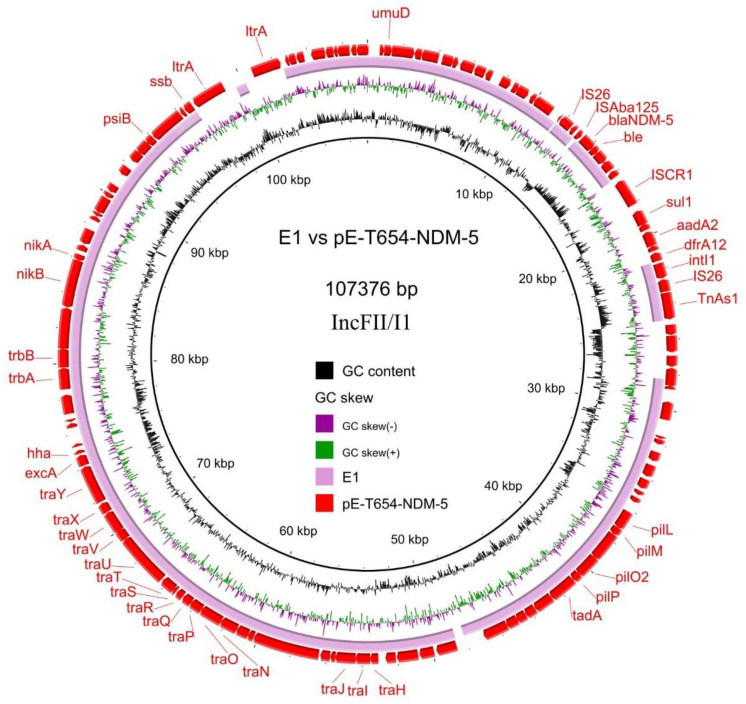
Circular alignments of the reference plasmid sequence (pE−T654−NDM−5, GenBank accession: CP090291) with homologous *bla*_NDM-5_−carrying contigs from the *E. coli* strain E1 in this study. Representative genes such as antimicrobial resistance genes, conjugation−associated genes and mobile genetic elements are labelled in the outermost circle.

**Figure 4 genes-14-00520-f004:**
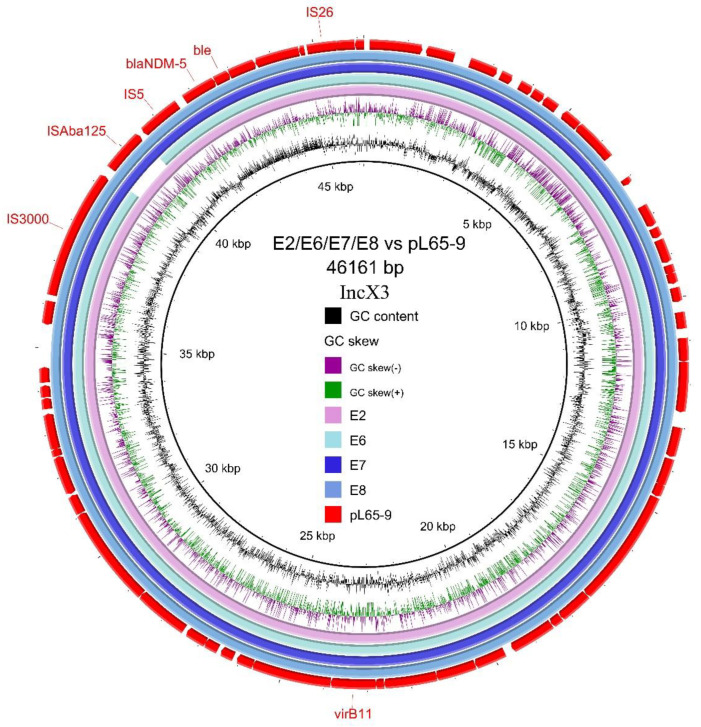
Circular alignments of the reference plasmid sequence (pL65−9, GenBank accession: CP034744) with homologous *bla*_NDM-5_-carrying contigs from the *E. coli* strains E2, E6, E7 and E8 in this study. Representative genes such as antimicrobial resistance genes and mobile genetic elements are labelled in the outermost circle.

**Figure 5 genes-14-00520-f005:**
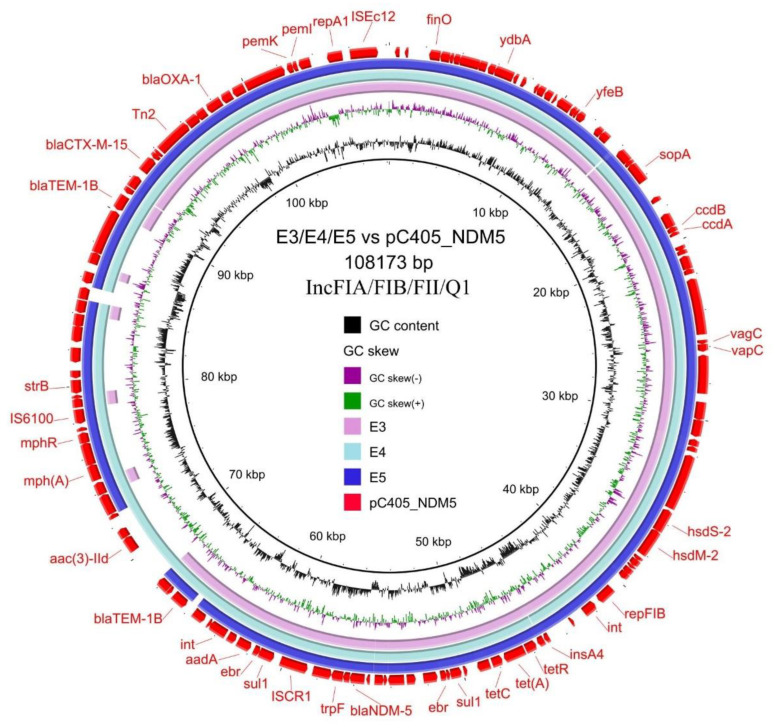
Circular alignments of the reference plasmid sequence (pC405−NDM5, GenBank accession: LC521844) with homologous *bla*_NDM-5_−carrying contigs from the *E. coli* strain E3, E4 and E5 in this study. Representative genes such as antimicrobial resistance genes, conjugation-associated genes and mobile genetic elements are labelled in the outermost circle.

**Table 1 genes-14-00520-t001:** Clinical information of the cohort.

Strain	Age *	Gender	Underlying Diseases	Clinical Department	Antibiotic Treatment	Outcome	Year of Isolation
E1	3 Y	male	acute myeloid leukemia	hematology	meropenem + amikacin	recovery	2019
E2	13 Y	male	acute lymphoblastic leukemia, post allo-HSCT	hematology	meropenem + amikacin, tigecycline	automatically discharged	2017
E3	1 M	female	neonatal sepsis, premature infant	neonatology	meropenem	recovery	2020
E4	9 D	female	necrotizing enterocolitis, neonatal sepsis	neonatology	meropenem	recovery	2020
E5	4 M	male	congenital atresia of biliary tract	gastroenterology	imipenem	recovery	2016
E6	11 Y	male	T lymphoblastic leukemia/lymphoma	hematology	tigecycline + amikacin	recovery	2020
E7	3 Y	female	acute lymphoblastic leukemia	hematology	meropenem + amikacin	recovery	2017
E8	12 D	male	neonatal sepsis, premature infant	neonatology	meropenem	recovery	2017

* Y, M and D in this column represent years, months and days, respectively.

**Table 2 genes-14-00520-t002:** MLST type and antibiotic resistance characteristics of *Escherichia coli* strains and their corresponding transconjugants.

Strains	Description	MLST	Serotype ^a^	Minimum Inhibitory Concentration (μg/mL) ^b^														Conjugation Efficiency ^c^
				AMP	SAM	ATM	SXT	CIP	TZP	GEN	FEP	CRO	CAZ	CTT	TOB	IPM	LVX	
E1	donor	ST38	O7:H8	≥32	≥32/16	2	≥16/304	1	64/4	≤1	≥64	≥64	≥64	≥64	≤1	≥16	1	NA
E2	donor	ST58	O?:H37	≥32	≥32/16	≤1	≥16/304	2	64/4	≥16	16	≥64	≥64	≥64	≥16	≥16	1	NA
E3	donor	ST410	O?:H9	≥32	≥32/16	≥64	≥16/304	≥4	≥128/4	≤1	≥64	≥64	≥64	≥64	≥16	4	≥8	NA
E4	donor	ST410	O?:H9	≥32	≥32/16	≥64	≥16/304	≥4	≥128/4	≤1	≥64	≥64	≥64	≥64	≥16	8	≥8	NA
E5	donor	ST410	O?:H9	≥32	≥32/16	4	≥16/304	0.5	≥128/4	≥16	≥64	≥64	≥64	≥64	8	≥16	1	NA
E6	donor	ST131	O25:H4	≥32	≥32/16	≥64	≥16/304	≥4	≥128/4	≥16	≥64	≥64	≥64	≥64	≥16	≥16	≥8	NA
E7	donor	ST361	O9:H30	≥32	≥32/16	16	≥16/304	≥4	≥128/4	≥16	≥64	≥64	≥64	≥64	≥16	≥16	≥8	NA
E8	donor	ST156	O11:H25	≥32	≥32/16	16	≥16/304	2	≥128/4	≤1	≥64	≥64	≥64	≥64	≤1	≥16	2	NA
E1-TC	transconjugant	ST80	O75:H7	≥32	≥32/16	≤1	≤1/19	≤0.25	64/4	≥16	16	≥64	≥64	≥64	≤1	≥16	0.5	2.5 × 10^−3^
E6-TC	transconjugant	ST80	O75:H7	≥32	≥32/16	16	≥16/304	1	≥128/4	≥16	≥64	≥64	≥64	≥64	≥16	16	1	2.9 × 10^−6^
E7-TC	transconjugant	ST80	O75:H7	≥32	≥32/16	16	≥16/304	≤0.25	≥128/4	≥16	≥64	≥64	≥64	≥64	≥16	≥16	0.5	3.5 × 10^−6^
E8-TC	transconjugant	ST80	O75:H7	≥32	≥32/16	≤1	≤1/19	≤0.25	64/4	≤1	16	≥64	≥64	≥64	≤1	≥16	0.5	1.6 × 10^−6^
EC600	recipient	ST80	O75:H7	16	8/4	≤1	≤1/19	≤0.25	≤4/4	≤1	≤1	≤1	≤1	≤4	≤1	≤1	0.5	NA

^a^ O? indicates that the O serotype of the strain was not typable using the current scheme. ^b^ Abbreviations: AMP, ampicillin; SAM, ampicillin–sulbactam; ATM, aztreonam; SXT, trimethoprim–sulfamethoxazole; CIP, ciprofloxacin; TZP, piperacillin–tazobactam; GEN, gentamicin; FEP, cefepime; CRO, ceftriaxone; CAZ, ceftazidime; CTT, cefotetan; TOB, tobramycin; IPM, imipenem; LVX, levofloxacin. ^c^ NA, not applicable.

## Data Availability

The assembled genome sequences of all *E. coli* strains from bloodstream infections in this study were deposited in the NBCI database under BioProject accession number PRJNA890582.

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
