# Peer review of "Characterization of NDM-5-Producing Escherichia coli Strains Isolated from Pediatric Patients with Bloodstream Infections in a Chinese Hospital"

_genes, 2023, doi:10.3390/genes14020520_

Round 1

Reviewer 1 Report

Referee Report

·        Figure 1 is very flu, I can not see some parameters, if possible should be revised.

·        For introduction section, the new articles were cited such as

Synthesis, spectroscopic characterization, crystal structure, DFT, molecular docking and in vitro antibacterial potential of novel quinoline derivatives, https://doi.org/10.1016/j.molstruc.2020.127940.

Syntheses of N-substituted benzimidazolone derivatives: DFT calculations, Hirshfeld surface analysis, molecular docking studies and antibacterial activities, https://doi.org/10.1016/j.molstruc.2019.127174.

·        All references should be controlled according to the Journal format.

MINOR REVISION

Author Response

"Please see the attachment

Reviewer 2 Report

Minor comments

1.      Please modify the sub-heading 2.1. Given heading is wrong as the paragraph is describing not only the collection and isolation but also the MALDI-TOF identification and confirmation of carbapenemase gene presence.

2.      In line 116, add “and” before congenital atresia.

3.      In line 120, write 7 and 1 in words.

4.      Rewrite and modify lines 143-145.

5.      Figure 2 ligand, correct axes to axis.

6.      Correct spellings of later in line number 237.

Major comments

Describe in one to two lines the isolation procedure of present study bacteria from patient’s samples. Also describe the medium used.

Discussion

The discussion portion is only focusing on the findings of present study. Kindly improve your discussion and compare the results with those reported in literature in detail.

Conclusion

Please describe in one to two lines that how this study can provide insight into the development of effective strategies for the control of untreatable BSIs caused by car-bapenem-resistant E. coli. Please suggest any approach or strategy.

Reviewer 3 Report

The manuscript entitled “Characterization of NDM-5-producing Escherichia coli strains recovered from bloodstream infections from pediatric patients in a Chinese hospital” is a significant epidemiological study on carbapenem-resistant E. coli strains producing the New Delhi metallo-β-lactamase-5, which has received widespread attention recently. However, the comments and suggestions listed below need to be addressed.

1.      As the manuscript reports the characterization of E. coli isolated from bloodstream infections, the bacteria should be referred to as "isolates" rather than "strains." I would suggest modifying the title to "Characterization of NDM-5-producing Escherichia coli Isolated from Pediatric Patients with Bloodstream Infections in a Chinese Hospital."

2.      Line 33: The authors are encouraged to highlight more of the significance of the study in the abstract.

3.      Lines 35-36: The keywords are missing. Please add them.

4.      Line 80: “β-lactam/β-lactamase combinations” should be “β-lactam/β-lactamase inhibitor combinations”

5.      Line 87: A reference citation and a brief description of the methodology for the filter-mating method are required.

6.      Lines 113-114: … which were 9 days, 12 days, 1 month, and 4 months, respectively. This sentence is confusing. Please remove “respectively” or the whole sentence. This information is mentioned in Table 1 anyway.

7.      In Table 1, it is notable that the number of blaNDM-5-producing E. coli isolates is higher in 2017 and 2020 than it was in 2016 and 2019. Are there any reasons for that? If so, please mention and discuss.

8.      Based on the CLSI 2020 data, the MIC breakpoint for resistance to ampicillin-sulbactam (SAM) is ≥ 32/16 µg/mL, to trimethoprim-sulfamethoxazole (SXT) is ≥ 4/76 µg/mL, and to piperacillin-tazobactam (TZP) is ≥ 128/4 µg/mL. Please present the data in Table 2 in a similar format. Why was the SXT tested at concentrations up to 320 µg/mL if the MIC breakpoint for resistance is ≥ 4/76 µg/mL?

9.      Lines 156-157: Please rephrase to remove ambiguity. E.g., Two strains (E1 and E2) carried the tetracycline resistance gene, tet(A), while four strains (E3, E4, E5, and E8) harbored the tetracycline resistance gene tet(B).

10.  Lines 195-199: This explanation is irrelevant to the “Results” section. Please move to the “Discussion” section.

11.  More depth is required in the discussion section to reflect the amount of work done in the manuscript and highlight the findings more. The authors are also encouraged to provide recommendations and suggest the next steps for future work.

12.  Some grammatical errors need correction.
